# Mindful self-focus–an interaction affecting Theory of Mind?

**Richard Wundrack** [ID]**\*, Jule Specht**

Department of Psychology, Humboldt-Universität zu Berlin, Berlin, Germany

\* richard.wundrack@hu-berlin.de

**Data Availability Statement:** All relevant data are within the paper and its Supporting information files.

**Funding:** This study was financially supported by the Deutsche Forschungsgemeinschaft (DFG, German Research Foundation) in the form of a

## Abstract

Is thinking about oneself helpful or harmful for understanding other people? The answer might depend on how a person thinks about themself. Mindfulness is one prominent construct that seems to affect the quality and content of a person's thoughts about themselves in the world. Thus, we hypothesize that the relationship between self-focus and Theory of Mind (ToM) is moderated by mindfulness. We evaluate our hypothesis with a large cross-sectional dataset ($N = 543$) of native and non-native German and English speakers using OLS and MM-estimated robust multiple regression analysis. We found a small but robust self-focus × mindfulness interaction effect on ToM so that there was a significant positive relation between self-focus and ToM for more mindful individuals and no significant relation for less mindful individuals. The findings support our hypothesis that mindfulness moderates the relationship between self-focus and ToM performance. We discuss the limitations and differences between the present study and previous findings.

## Introduction

Is thinking about oneself helpful or harmful for understanding other people? *Self-focus* is the tendency to attend to one's own thoughts, feelings, and intentions [1,2], while *Theory of Mind* (ToM) is the ability to infer these in other people [3,4]. Intuitively, the tendency to think about oneself should bias inferences about other people in an egocentric manner [5]. *Objective self-awareness theory*, however, posits that self-focus actually reduces this bias because it suggests self-focus is taking a third-person perspective at oneself [6–8]. A third possibility is that self-focus can be helpful or harmful depending on *how* one thinks about oneself.

This third option is derived from the *meta-construct model* [2,9]. Self-focus is part of normal psychological functioning, yet historically it has often been considered regarding excessive self-focus associated with negative affect, anxiety, depression, and virtually every mental disorder [10–12]. Within this context, Ingram suggested that one should distinguish between the pervasive *process* of self-focus and its specific *content* or *quality* when evaluating its role.

Here, we explore this idea by investigating whether the relationship between *self-focus* and *ToM* performance is moderated by *mindfulness*. *Mindfulness* is the tendency to be conscious of what is going on in the present moment within oneself and in one's surroundings including other people [13,14]. Thus, mindfulness is an ideal candidate for a moderator affecting the

grant (491192747). This study was also financially supported by the Open Access Publication Fund of Humboldt-Universität zu Berlin. The funders had no role in study design, data collection and analysis, decision to publish, or preparation of the manuscript.

**Competing interests:** The authors have declared that no competing interests exist.

content and quality of a person's attention, independent of whether it is focused on oneself, others, or something else. Indeed, previous research has shown that reflective or mindful self-focus has many benefits for psychological functioning making mindfulness a likely moderator for our hypothesis [15–20].

## The outcome: Theory of Mind (ToM)

In real life, the ability to reason about other people's mental states requires the consideration of past, present, general, and occasion-specific information about people and social situations [21]. Additionally, one's own mental state regarding some context may provide valuable insight into other people's perspectives. Often, what is shared already explains quite a lot [22]. However, one can also be mistaken to project one's mental state onto other people or believe they are likeminded, that is *egocentric bias* and *false consensus belief*, respectively [4,23,24], the crux for successful ToM is appropriately differentiating between oneself and another person, that is *self–other distinction* [25].

Notably, bias and accuracy are not necessarily opposites; in the right circumstances, bias can facilitate accuracy because bias allows for robust predictions under uncertainty [26,27]. In other words, a person can be right for the "wrong" reason like when grounding inferences about other people's mental states in their own mental state rather than information about the other person. This is important because most ToM tasks measure either accuracy or bias but not both. Thus, they actually cannot answer whether in real life more egocentric participants will be less accurate or *vice versa*. Previous research on self-focus and ToM has mostly employed measures of egocentric bias, while here an accuracy measure was used.

## The predictor: Self-focus

As mentioned, the tendency to focus on one's own mental states has often been considered from a psychopathological perspective. Much research differentiated between private and public, positive and negative, or reflective and ruminative self-focus [11] reinforcing the idea that the role of self-focus depends on its quality [2,9].

Nevertheless, most research focuses on the main effects of some kind of self-focus. For example, some studies have suggested a negative effect [5,28], and others have suggested a positive effect of self-focus on ToM [29,30]. With tasks like (a) writing an E on one's forehead, (b) judging how a third party would interpret a sentence the participant knows to be meant sarcastically, or (c) estimating how many peers share one's preferences, these studies examine the role of self-focus for egocentric bias but not for ToM accuracy.

We found only one study that employed a ToM accuracy measure, specifically *emotion recognition* [31]. Therein, accuracy was based on the comparison of participant ratings of the emotions conveyed in different video clips and the actors' self-ratings of their enacted emotions. However, the study investigated the role of *self-referential processing* which concerns the superior recall of information that has previously been related to oneself as compared to information that has not been related to oneself [32]. In contrast, self-focus is about the act and tendency to relate information to oneself in the first place. As such, self-focus is a prerequisite of self-referential processing. The study found that participants who better retrieved self-related information were also more accurate in judging other's emotions.

On grounds of research on objective self-awareness theory [6,7,33], self-focus seems to be positively related to ToM. Arguably, it facilitates taking a third-person perspective on one's own perspective. In other words, it should help with appropriate self–other distinction to account for otherwise misplaced egocentric bias. This leads to our first cautious hypothesis:

*(H1) If there is a main effect of self-focus on ToM at all, it is probably positive.*

## The moderator: Mindfulness

Being aware of the present moment is relevant to both inward self-focus and outward ToM. Quite generally, research suggests a positive relation of mindfulness and most social cognitive abilities [34,35]. Disputed is mainly the underlying mechanism. Suggestions include among others that mindfulness improves ToM (a) by increasing self-knowledge or self-compassion [36–38], (b) by simply motivating a person to engage more in ToM [39], or (c) decreasing egocentric biases directly or indirectly through changes in affect [40–42]. Thus, we hypothesize:

*(H2) There is a positive main effect of mindfulness on ToM.*

We chose mindfulness as a likely moderator for self-focus because by definition mindfulness shapes how we relate to ourselves in the world and thus should determine the content or quality of our self-focus, for example, as being directed at one's present thoughts and feelings and being non-judgmental about them. A range of distinct adaptive properties have been discerned for reflective or mindful self-focus on mood and psychological functioning [15,17,18,20]. For example, it has recently been shown that paranoid thinking is maintained by ruminative self-focus but reduced by mindful self-focus [16].

Regarding ToM specifically, however, evidence is sparse. We found only one study that tested how the effect of self-focus on ToM was moderated but therein, the moderator was negative affect [43]. After inducing states of shameful, guilty, or neutral self-focus, the researchers asked participants to judge how sarcastic an uninformed third person would interpret a message praising a poor restaurant experience. Ashamed individuals expected a more sarcastic interpretation and guilt-ridden individuals expected a less sarcastic interpretation than individuals in the neutral condition. This illustrates that the direction of a person's egocentric bias can change depending on the (affective) quality of self-focus. Taken together with Ingram's suggestion [2], this motivates our central hypothesis:

*(H3) Mindfulness positively moderates the relation of self-focus and ToM.*

## Control variables

**Negative affect.** The tendency or state of experiencing negatively valanced feelings [44,45] has come up multiple times. Negative affect is a broad construct comprising different feelings that in themselves serve widely different socio-psychological functions [46,47]. Different affective states can play different roles for both thinking style and thought content depending on the context and the object of affect attributions [48–51].

Thus, it should not be surprising that relation of negative affect to self-focus and ToM is not clear cut [2,11,12,15,17–20]. Case in point are the study on the role of shameful and guilt-ridden self-focus for egocentric bias [43] and a similar study suggesting that states of anxiety and surprise, but not anger or disgust drive individuals to rely more on their own perspective [52]. In contrast, the relation between negative affect and mindfulness clearly seems to be negative with an $r$ = -.39 [53].

We consider negative affect an important control variable because it seems to be related to both self-focus and ToM–though at the domain level the direction remains unclear.

**Other influential variables.** There are several additional variables we consider: age, years of education, gender, participation language, language nativity, ToM task attention, and study participation duration. Although we do not expect either of them to drastically change the hypothesized relations, they are likely candidates to explain some of the variance and provide

some context for the interpretation of the effects of interest [54]: We expect ToM performance to be negatively related to age [55] but positively to years of education [56], and to be worse in male participants [57], in non-native speakers [58], and inattentive participants. Study participation duration may introduce noise to the data but should not significantly affect ToM performance in a particular direction or affect its relation to self-focus and mindfulness as both constructs are rather traits than states according to the understanding underlying the used measurements (compare *Materials*).

## Materials and methods

### Participants

The study was approved by our department's ethics committee (proposal number 2020–01). Between mid-February 2020 and mid-April 2020 $N$ = 584 individuals were recruited by different means of on- and offline advertisement. Participant gave written informed consent. Native and non-native individuals above the age of 18 could take part in German or in English ($N$ = 291 German natives, $N$ = 53 German non-natives, $N$ = 75 English natives, and $N$ = 162 English non-natives). Compensation comprised personalized feedback, a 50 €-raffle per 100 participants, and study participation credit for local psychology undergraduates ($N$ = 44). Participants were fairly international, being native to 61 different countries while residing in 37 different countries–though the majority were either German ($N$ = 298) or residing in Germany at the time of the study ($N$ = 421). Participants identified mostly as females ($N$ = 419), were largely in their late twenties (median age = 29, range 18–88), and highly educated (highest degree achieved at a university ($N$ = 353)). In summary, the sample was WEIRD [59,60].

### Materials

**Self-focus.** Self-focus was measured with the self-focus sentence completion task (SFSC) [1] which requires subjects to finish 30 open-ended sentences prompting responses concerning themselves or others, e. g. "If only I could . . .". Each half-sentence response was coded by three raters according to the coding scheme suggested by Exner across his six categories: "egocentric" (self-focused, e.g. ". . . live my life freely."), "egocentric and negative" (e.g. ". . . end my life."), "allocentric" (other-focused, e.g. ". . . help my sister."), "allocentric and emotional" (e.g. ". . . stop hating my father for what he has done."), "both" (self- and other-focused, e.g. ". . . repair my relationship with my mother."), and "other" for answers that do not relate to a person or are too short (e.g. ". . . fly"). We followed modern research practice and evaluated the SFSC based on the count of "egocentric" and the "egocentric and negative" responses [12,61,62]. We divided their sum by the number of raters and the number of SFSC items to get a ratio of self-focus that is as unbiased as possible by the coding variability of an individual rater. The three raters were psychology students previously trained on a pilot sample ($N$ = 73). Interrater reliability was *Fleiss'* $\kappa$ = 0.695 (p < 0.001, 95% CI [0.692; 0.697]; [63]).

**Mindfulness.** Trait mindfulness was assessed using the Mindfulness, Attention, and Awareness Scale (MAAS) [14,64]. The MAAS is a popular 15-item frequency measure of dispositional mindfulness, including receptive awareness of, and attention to what takes place in the present moment (e. g. "I snack without being aware that I am eating."). All items are reverse-coded and rated on a 6-point Likert scale ranging from "almost always" (1) to "almost never" (6). Measurement reliability was $\alpha$ = .84 and $\lambda_6$ = .84 [65,66].

**ToM.** Here, we used the Double Movie for Assessment of Social Cognition–Multiple Choice (DMASC-MC) [67,68]. Throughout a 15-minute short movie, the DMASC-MC requires participants to answer 44 items on the thoughts, feelings, and intentions of four characters who spend an evening together (e. g. "Why did Michael say that?"). Each time,

participants selected one from four multiple choice options indicating what they think was true, which was coded as based on the DMASC-MC as "mentalized appropriately", "too much", "too little", or "not at all". We also included five attention checks inquiring which topics have been extensively discussed among the characters (e. g. what to cook for dinner). Our implementation of the DMASC-MC automatically jumped to the next video sequence as soon as participants selected an answer.

**Negative affect.**   Negative trait affect was measured alongside positive affect (not considered here) with the International Positive And Negative Affect Schedule Short Form (I-PANAS-SF) [69] which measures negative trait affect through subjects' self-rating with five items ('upset', 'hostile', 'ashamed', 'nervous', 'afraid') on a 5-point Likert scale ranging from "not at all" (1) to "extremely" (5). Measurement reliability was $\alpha$ = .79 and $\lambda_6$ = .77.

**Other covariates.**   Among others, participation language (dichotomous: German/English), participation language nativity (dichotomous: native/non-native), gender (dichotomous: male/female), age (continuous), years of education (continuous), and ToM task attention (five multiple choice control items), and study duration (time-stamped) were assessed.

## Procedure

Data collection was done in formr [70,71]. After being informed about the purpose of the study and agreeing to its terms and conditions, participants answered to the SFSC, the I-PANAS-SF, the MAAS, the BFI-2-S (not considered here) [72,73], another pilot questionnaire on the variability in Big Five trait expression (not considered here). Subsequently, participants completed the DMASC-MC and provided demographic information before finishing the study by choosing their means of compensation.

Except for the demographic and compensatory information, responses were mandatory. Due to the estimated length of the study (ca. 1h), participants were invited to take breaks between the tasks. In combination with the lack of a preset study expiration time, this led some individuals to spread their participation over a couple of hours or even days. The median study duration excluding study consent and compensation was 62 min with $N$ = 517 subjects participating within 2h, $N$ = 56 more participated within 24h, and $N$ = 11 taking multiple days up to one week.

## Data analysis

**Data preparation.**   We included all participants who got as far as fully completing the DMASC-MC ($N$ = 584) and correctly answered at least 4 out of 5 attention check items during that task (out $N$ = 41). We did not exclude participants for any other reason. Missing data for years of education ($N$ = 48), gender ($N$ = 15), age ($N$ = 10), language nativity ($N$ = 3), and study duration ($N$ = 1) were imputed based on the variables included in the joined model (cf. section: Multiple Regression Analysis) using predictive mean matching for the continuous variables [74] and logistic regression for categorical variables [75]. We deemed a single imputation without variance estimation sufficient because it only concerned control variables.

**Equivalence testing.**   Descriptive statistics include equivalence testing following the *two one-sided test* procedure (TOST) [76,77]. This allowed us to judge whether small but according to *null hypothesis significance testing* (NHST) significant differences ($p < .05$) between the German and English subsamples were nevertheless statistically equivalent to zero based on the statistically necessitated threshold of the *smallest reliably detectable effect size* with a 90% confidence interval.

**Multiple regression analysis.**   ToM performance was predicted through multiple linear regression. We ran an interaction model with self-focus, mindfulness, and the self-

focus × mindfulness interaction, a covariates model including the following control variables: language (English vs German), language nativity (non-native vs native), gender (male vs female), correct attention control items (4/5 vs. 5/5), age, years of education, negative affect, and study duration. All continuous variables were z-standardized to better meet OLS assumptions, to prevent multicollinearity, and for better comparability across variables [78]. For all categorical variables, contrasts were set using weighted effect coding to account for their imbalanced distribution [79]. Finally, we ran the joined model including all predictors from the interaction and the covariates model.

For the central interaction effect model, we determined the smallest reliably detectable effect size through sensitivity power analysis (given 3 predictors, $\alpha = .05$, $power = .95$, $N = 543$) to be Cohen's $f^2 = .032$ for the whole model and a partial $f^2 = .024$ for one of three predictors [80]. Furthermore, we ran each model as an OLS and an MM-estimated robust regression model [81] to judge results independent of parametric assumptions.

**Data and analysis access and software.** With exception of the sensitivity power analysis done in G*Power version 3.1 [82], data analysis was entirely done in R version 4.0.2 [83] through R Studio version 1.4.1103 [83] using the following packages: broom [84], car [85], clickR [86], here [87], interactions [88,89,90], interplot [90], lmtest [91], MASS [92], mice [93], misty [94], performance [95], psych [96], sensemakr [97], sjmisc [98], tidyverse [99], TOSTER [76], and wec [79,100].

## Results

The pseudonymized and scale-aggregated data and the analysis script are publicly available through the Open Science Framework: (https://osf.io/yneu7/). Descriptive statistics for all variables pooled and broken down by participation language are presented in Table 1. The

**Table 1. Descriptive statistics.**

|  |  | Total | | German | | English | | Statistical Equivalence | |
|---|---|---|---|---|---|---|---|---|---|
|  |  | $N = 543$ | | $N = 334$ (61.51%) | | $N = 209$ (38.49%) | | | |
| **Continuous variables** | | **M** | **SD** | **M** | **SD** | **M** | **SD** | **TOST** | **NHST** |
| ToM performance | | 34.29 | 4.04 | 34.56 | 4.08 | 33.87 | 3.94 | * | ns |
| Age | | 32.19 | 11.57 | 32.80 | 12.87 | 31.22 | 9.05 | * | ns |
| Years of education | | 15.31 | 4.70 | 15.67 | 4.43 | 14.75 | 5.07 | * | * |
| Negative affect | | 2.03 | 0.76 | 1.85 | 0.68 | 2.32 | 0.78 | ns | * |
| Study duration | | 145.75 | 550.89 | 165.95 | 672.77 | 113.46 | 253.90 | * | ns |
| Mindfulness | | 3.94 | 0.71 | 3.99 | 0.70 | 3.88 | 0.72 | * | ns |
| Self-focus | | 0.34 | 0.09 | 0.35 | 0.09 | 0.32 | 0.09 | ns | * |
| **Categorical variables** | | **N** | **%** | **N** | **%** | **N** | **%** | **TOST** | **NHST** |
| Nativity | - native | 352 | 64.48 | 286 | 85.63 | 66 | 31.58 | ns | * |
|  | - non-native | 191 | 35.17 | 48 | 14.37 | 143 | 68.48 | | |
| Gender | - female | 402 | 74.03 | 254 | 76.05 | 148 | 70.81 | * | ns |
|  | - male | 141 | 25.97 | 80 | 23.95 | 61 | 29.19 | | |
| ToM attention | - 5/5 | 411 | 75.69 | 275 | 82.34 | 136 | 66.51 | ns | * |
|  | - 4/5 | 132 | 24.31 | 59 | 17.66 | 73 | 33.49 | | |

Listed are the pooled and language-group specific (a) means (M) and standard deviations (SD) of the continuous variables and (b) absolute and relative values for the categorical variables after imputation. Furthermore, the overview provides significant (*; at p < .05) and non-significant (ns) null hypothesis significant testing (NHST) and two one-sided test procedure (TOST) results comparing the statistical equivalence of the German and English subsample. A significant NHST result indicates the difference between the German and English subsample was statistically different from zero and more importantly a significant TOST result indicates the difference was statistically *equivalent* to zero.

**Table 2. Correlation matrix.**

| | | | continuous | | | | | | | categorical | | |
|---|---|---|---|---|---|---|---|---|---|---|---|---|
| continuous | **1** | **ToM performance** | **1** | **2** | **3** | **4** | **5** | **6** | **7** | **8** | **9** | **10** |
| | **2** | **Age** | -.11 | | | | | | | | | |
| | **3** | **Years of education** | .13 | .13 | | | | | | | | |
| | **4** | **Negative affect** | -.08 | -.11 | -.03 | | | | | | | |
| | **5** | **Study duration** | .01 | .08 | .03 | -.03 | | | | | | |
| | **6** | **Mindfulness** | -.01 | .18 | .03 | -.32 | -.03 | | | | | |
| | **7** | **Self-focus** | .06 | -.16 | .01 | .10 | .00 | -.10 | | | | |
| categorical | **8** | **Language** | .11 | .08 | .12 | -.39 | .06 | .10 | .19 | | | |
| | **9** | **Nativity** | .25 | .11 | .07 | -.25 | -.02 | .03 | .17 | .55 | | |
| | **10** | **Gender** | -.13 | .24 | -.02 | -.02 | -.04 | .09 | -.05 | .04 | .00 | |
| | **11** | **ToM attention** | .19 | -.08 | .07 | .07 | .05 | .05 | .08 | .19 | .12 | .06 |

Continuous–continuous correlations (top left) have been computed as Pearson correlations; categorical–categorical correlations (bottom right) as bias-corrected Cramer's V; and continuous–categorical correlations (bottom left) as biserial correlations. Note that correlation coefficients cannot be directly compared across combinations of variable types due to different underlying assumptions.

German subsample was slightly more attentive during the ToM task, reported less negative affect, was slightly more self-focused, contained fewer non-native speakers than the English subsample, and took an average 40 min longer to complete the study. A correlation matrix is provided in Table 2. Table 3 details the results of the regression analyses. In particular, we provide the OLS and the respective MM-estimated robust counterpart of the covariates model (($F_{(8;534)} = 7.005$, $p < .001$, adj. $R^2 < 0.081$, Cohen's $f^2 = .104$); ($\sigma^{residual}(534) = 0.933$, Cohen's $f^2 = .103$)), the interaction model (($F_{(3;539)} = 3.124$, $p = .026$, adj. $R^2 = .012$, Cohen's $f^2 = .017$); ($\sigma^{residual}(539) = 0.988$, Cohen's $f^2 = .017$)), and the joined model (($F_{(11;531)} = 5.763$, $p < .001$, adj. $R^2 = .088$, Cohen's $f^2 = .119$); ($\sigma^{residual}(531) = 0.92$, Cohen's $f^2 = .117$)). Notably, the self-focus × mindfulness interaction was significant in the OLS and the robust interaction models and the covariates models. Furthermore, comparing models indicated that age, years of education, language nativity, and ToM task attention were significantly related to ToM performance but did not account for the variance explained by the self-focus × mindfulness interaction. Negative affect and participation language were only significant in the robust model. Lastly, regression diagnostics as well as the near-perfect correlation $r = .99$ between the residuals of the OLS models and their MM-estimated robust counterparts suggest any violations of OLS assumptions were negligible [101].

## Discussion

We set out asking whether thinking about oneself is helpful or harmful for ToM performance. Reviewing the sparse and mixed literature, we found Ingram's theory [2] most compelling that the answer may depend on the specific *content* or *quality* a person's self-focus can take. We considered mindfulness a psychological construct that should affect the content or quality of self-focus because mindfulness specifies towards what and how a person focuses their attention. Thus, we explored the idea whether the relationship between *self-focus* and *ToM* performance is moderated by *mindfulness*. Overall, our results are in line with Ingram's idea findng different effects of self-focus on ToM performance depending on a person's level of mindfulness.

Most importantly, we found support for our central hypothesis (H3) that there is a moderation effect of mindfulness on the relation between self-focus and ToM. The moderation effect

**Table 3. Regression models.**

| Models [AIC; BIC] | | Covariates Model [1506; 1549] | | | | | | Joined Model [1505; 1561] | | | | | | Interaction Model [1541; 1562] | | | | | |
|---|---|---|---|---|---|---|---|---|---|---|---|---|---|---|---|---|---|---|---|
| Variable | Regres. | β | SE | 95% CI | | p | f^2 | β | SE | 95% CI | | p | f^2 | β | SE | 95% CI | | p | f^2 |
| | | | | LL | UL | | | | | LL | UL | | | | | LL | UL | | |
| **Intercept** | OLS | .161 | .055 | .053 | .269 | .003 | .016 | .169 | .055 | .061 | .277 | .002 | .018 | .012 | .043 | -.072 | .096 | .778 | .000 |
| | *robust* | *.204* | *.055* | *.092* | *.313* | *< .001* | | *.214* | *.054* | *.106* | *.318* | *< .001* | | *.076* | *.042* | *-.011* | *.16* | *.073* | |
| **Age** | OLS | -.123 | .043 | -.207 | -.039 | .004 | .015 | -.109 | .044 | -.195 | -.022 | .014 | .012 | | | | | | |
| | *robust* | *-.111* | *.043* | *-.222* | *-.015* | *.011* | | *-.088* | *.044* | *-.204* | *.004* | *.05* | | | | | | | |
| **Years of education** | OLS | .136 | .042 | .054 | .218 | .001 | .020 | .138 | .042 | .056 | .22 | .001 | .021 | | | | | | |
| | *robust* | *.146* | *.041* | *.07* | *.233* | *< .001* | | *.149* | *.041* | *.067* | *.233* | *< .001* | | | | | | | |
| **Negative affect** | OLS | -.068 | .043 | -.153 | .017 | .118 | .005 | -.082 | .046 | -.173 | .008 | .074 | .006 | | | | | | |
| | *robust* | *-.079* | *.043* | *-.161* | *-.003* | *.066* | | *-.102* | *.045* | *-.184* | *-.006* | *.026* | | | | | | | |
| **Study Duration** | OLS | .016 | .041 | -.065 | .098 | .693 | .000 | .014 | .041 | -.067 | .096 | .728 | .000 | | | | | | |
| | *robust* | *.008* | *.04* | *-.043* | *.092* | *.841* | | *.006* | *.039* | *-.034* | *.096* | *.875* | | | | | | | |
| **Gender** [male] | OLS | -.134 | .071 | -.273 | .005 | .059 | .007 | -.122 | .071 | -.261 | .017 | .085 | .006 | | | | | | |
| | *robust* | *-.099* | *.071* | *-.26* | *.048* | *.161* | | *-.092* | *.07* | *-.237* | *.053* | *.192* | | | | | | | |
| **Language** [English] | OLS | .114 | .065 | -.014 | .242 | .081 | .006 | .125 | .066 | -.004 | .255 | .057 | .007 | | | | | | |
| | *robust* | *.122* | *.065* | *-.023* | *.262* | *.063* | | *.143* | *.066* | *-.001* | *.299* | *.03* | | | | | | | |
| **Nativity** [non-native] | OLS | -.459 | .104 | -.663 | -.255 | < .001 | .037 | -.449 | .104 | -.653 | -.245 | < .001 | .035 | | | | | | |
| | *robust* | *-.421* | *.104* | *-.679* | *-.207* | *< .001* | | *-.421* | *.104* | *-.665* | *-.191* | *< .001* | | | | | | | |
| **ToM attention** [4 out of 5] | OLS | -.185 | .074 | -.331 | -.038 | .013 | .012 | -.187 | .074 | -.333 | -.041 | .012 | .012 | | | | | | |
| | *robust* | *-.202* | *.075* | *-.361* | *-.036* | *.007* | | *-.211* | *.074* | *-.384* | *-.046* | *.005* | | | | | | | |
| **Self-focus** | OLS | | | | | | | .044 | .044 | -.041 | .13 | .31 | .002 | .081 | .044 | -.004 | .167 | .063 | .006 |
| | *robust* | | | | | | | *.06* | *.043* | *-.031* | *.137* | *.166* | | *.079* | *.043* | *0* | *.158* | *.066* | |
| **Mindfulness** | OLS | | | | | | | -.008 | .044 | -.094 | .079 | .863 | .000 | .004 | .043 | -.08 | .089 | .921 | .000 |
| | *robust* | | | | | | | *-.009* | *.044* | *-.117* | *.094* | *.838* | | *.029* | *.043* | *-.073* | *.128* | *.505* | |
| **Self-focus × Mindfulness** | OLS | | | | | | | **.113** | **.044** | **.027** | **.199** | **.01** | **.012** | **.122** | **.045** | **.033** | **.211** | **.007** | **.014** |
| | *robust* | | | | | | | ***.128*** | ***.044*** | ***.03*** | ***.222*** | ***.004*** | | ***.117*** | ***.045*** | ***.025*** | ***.216*** | ***.009*** | |

All continuous variables have been z-standardized and all categorical variables have been weighted effect coded. Note: Significant findings p > .05 are highlighted in bold.

was weak but significant in the OLS ($β = .122$, $p = .007$) and the robust ($β = .117$, $p = .009$) interaction model. The relation between self-focus and ToM performance became positive when the mean level of mindfulness was exceeded (Figs 1 and 2).

Notably, the moderation effect remained about the same strength even when controlling for covariates in the OLS ($β = .113$, $p = .01$) and the robust model ($β = .128$, $p = .004$). The interaction effect had about the same strength as age ($β = -.109$, $p = .014$) and years of education ($β = .138$, $p = .001$), while attention to the ToM task ($β = -.187$, $p < .012$) was almost twice as strong. Interestingly, language nativity ($β = -.449$, $p < .001$) was still four times as strongly related to ToM performance although participants clearly understood enough about the non-mental content of the conversations in the ToM task to pass the attention check.

A crucial question is whether we should care about the moderation [77,102,103,104] because its effect size was smaller (partial $f^2 = .014$) than the smallest reliably detectable effect size as determined by our sensitivity power analysis (partial $f^2 = .024$)–which was coincidentally close to Cohen's (1988) benchmark for small effects. It suggests that in the long run our finding may be associated with a Type I error rate exceeding the targeted 5% to some extent. By the same reasoning, however, here age and years of education were also unreliable

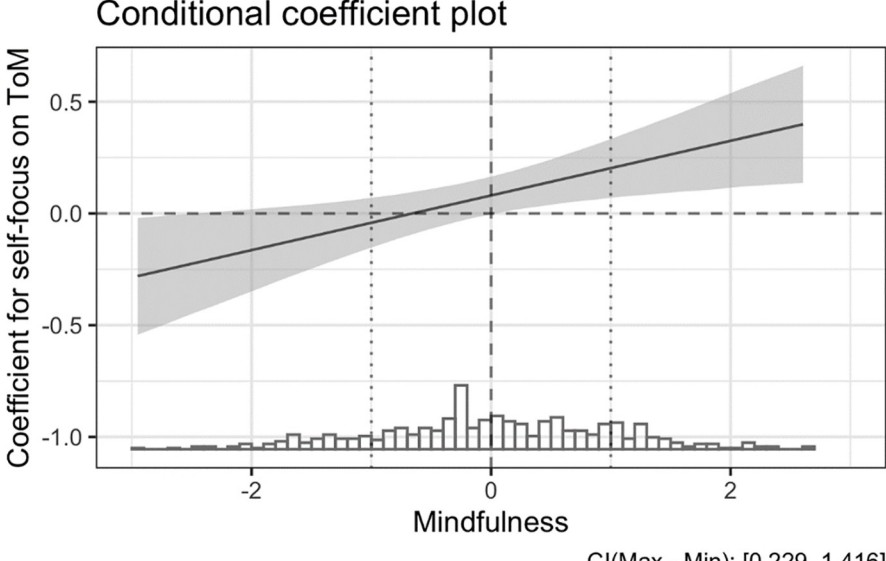

**Fig 1. Conditional coefficient plot.** Based on the OLS interaction effect model without control variables. Dotted vertical lines indicate -/+ 1 SD for mindfulness, the dashed vertical line indicates the mean. The plot shows how the relationship between self-focus and ToM performance, the coefficient, changes from negative to positive as the mindfulness level continuously increases.

predictors of ToM although their significant relation with ToM has been repeatedly shown. Still, future studies should account for this by increasing their sample size.

Nevertheless, we believe that our results are informative given this study is the first of its kind relating self-focus and ToM accuracy instead of egocentric bias in a sample this large. First, our

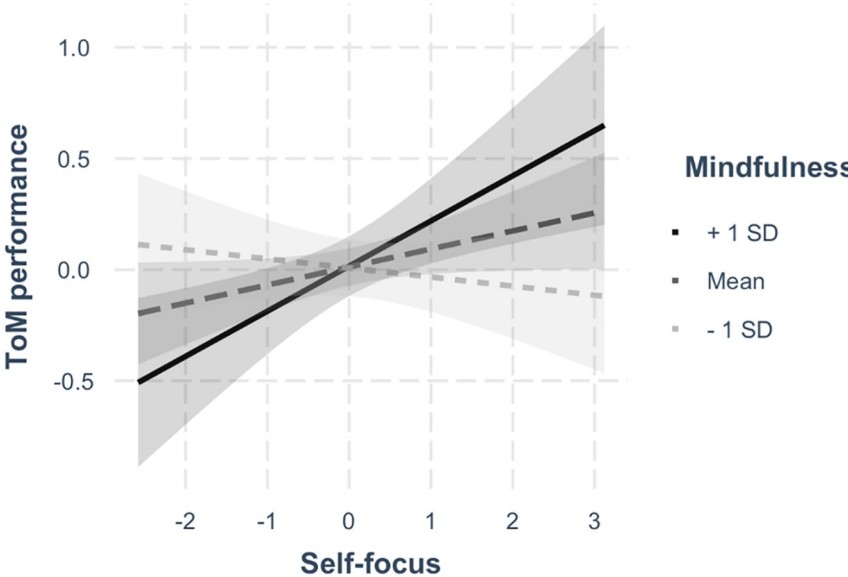

**Fig 2. Interaction effect plot.** Based on the OLS interaction effect model without control variables. The plot shows how the relationship between self-focus and ToM performance is different for high, average, and low levels of mindfulness, i.e., +1 SD, mean, and -1 SD, respectively. It is a discrete visualization of the continuous relationship depicted in Fig 1.

finding suggests that even if self-focus affects egocentric bias as suggested by previous findings, this may not directly translate into better or worse ToM. A speculative reason may be a "tradeoff of egocentrism" between a person's own perspective being a source of bias and a source of information when reasoning about other people's mental states. As argued earlier, bias is usually considered detrimental to accuracy but can be advantageous given noisy information. It may be that mindfully self-focused individuals optimize this trade-off, while absentmindedly self-focused individuals fall short of recognizing their bias or the informativeness of their own perspective.

A second justification for the small but robust moderation effect might be that inferring other people's minds is a complex task involving a person's immediate mental state and other situational circumstances like the availability of more target-specific information. From this perspective, even the small interaction effect of two trait-like constructs (self-focus and mindfulness) may seem quite reasonable.

Our first and second hypotheses concerning the positive main effects of self-focus and mindfulness on ToM performance were not supported. However, together with the significant mediation effect, this may only strengthen our main claim that the role of self-focus for ToM is dependent on the quality or content of self-focus like whether self-focus is mindful or absent-minded. The finding does not support objective self-awareness theory but it does not directly oppose it either because the SFSC arguably assesses a dispositional form of self-focus, whereas objective self-awareness theory is concerned with the role of state self-focus.

## Limitations

Previous research on self-focus often relied on (quasi-)experimental designs in smaller samples often inducing different state levels of self-focus while measuring egocentric bias. We analyzed a large cross-sectional dataset including–depending on one's interpretation of what the SFSC measures–trait self-focus and a ToM accuracy measure.

The relationships between variables were generally weak with two key measures suffering methodological criticism: although in use for a long time, the SFSC's validity and reliability are questionable as there has been no formal validation against other measures of self-focus [1] and the MAAS items ask exclusively about absentminded behavior but the absence of absent-mindedness might not equal mindfulness [104]. Moreover, our data might have been quite noisy: first, the study's overall procedure might have taken too long and been too demanding for an online study for which a distraction-free environment cannot be guaranteed; second, collapsing across German and English natives and non-natives might make the observations more heterogeneous without making the findings more generalizable.

## Conclusion

We hypothesized that the role of self-focus on ToM performance depends on a person's level of mindfulness so that focusing on oneself, may hinder or facilitate accurate ToM or not. We found a small but robust and significant interaction effect of self-focus and mindfulness according to which there is a positive effect on ToM performance for mindfully self-focused individuals but not for absentminded individuals. Thus, our results provide initial evidence for the idea that ToM performance is differentially influenced by different qualities of self-focus. Future research is needed to investigate the exact mechanisms at work in this relationship.

## Supporting information

**S1 File.**
(ZIP)

**S2 File.**
(ZIP)

# Acknowledgments

The authors are most grateful for the supportive work of their student assistants I. Gharago-zlou, L. Göbel, A. Kayumi, P. Maschke, and P. Schumann.

# Author Contributions

**Conceptualization:** Richard Wundrack, Jule Specht.

**Data curation:** Richard Wundrack.

**Formal analysis:** Richard Wundrack.

**Methodology:** Richard Wundrack.

**Project administration:** Richard Wundrack, Jule Specht.

**Supervision:** Jule Specht.

**Validation:** Richard Wundrack.

**Visualization:** Richard Wundrack.

**Writing – original draft:** Richard Wundrack.

**Writing – review & editing:** Richard Wundrack, Jule Specht.

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
