## [Decision Letter · Decision Letter 0]

22 Jul 2022

PONE-D-22-04751Mindful Self-Focus–An Interaction Affecting Theory of Mind (ToM)?PLOS ONE

Dear Dr. Wundrack,

Thank you for submitting your manuscript to PLOS ONE. After careful consideration, we feel that it has merit but does not fully meet PLOS ONE’s publication criteria as it currently stands. Therefore, we invite you to submit a revised version of the manuscript that addresses the points raised during the review process.

We look forward to receiving your revised manuscript.

Kind regards,

Yi-Yuan Tang

Academic Editor

PLOS ONE

Journal Requirements:

The author(s) received no specific funding for this work. The first author was funded by the Friedrich-Ebert-Foundation as a Phd student.

6. Please ensure that you include a title page within your main document. You should list all authors and all affiliations as per our author instructions and clearly indicate the corresponding author.

Additional Editor Comments:

Dear Dr. Richard Wundrack,

We finally got two reviews back and please revise and resubmit your MS based on reviewers' comments, thanks!

Reviewers' comments:

Reviewer's Responses to Questions

**Comments to the Author**

1. Is the manuscript technically sound, and do the data support the conclusions?

Reviewer #1: Yes

Reviewer #2: Yes

2. Has the statistical analysis been performed appropriately and rigorously? 

Reviewer #1: Yes

Reviewer #2: Yes

3. Have the authors made all data underlying the findings in their manuscript fully available?

Reviewer #1: Yes

Reviewer #2: Yes

4. Is the manuscript presented in an intelligible fashion and written in standard English?

Reviewer #1: No

Reviewer #2: Yes

5. Review Comments to the Author

Reviewer #1: In the study of the relationship between self focus and theory of mind (ToM), it is theoretically feasible to include mindfulness level as a regulating variable.

The following problems exist:

1. The measured ToM in this paper is obtained through the coding of the experimenter, and the individual judgment standard of the coder will have an unstable impact on the results.

2. Among the subjects participating in the experiment, a small number of subjects complete the cycle too long, which will lead to different psychological states of the same subject at different times, which will affect the experimental results (on page 8).

3. Before putting forward the hypothesis, I suggest the author introduce the background or reason for choosing mindfulness as the regulating variable? (In abstract)

4. On the second page, I suggest the author outline the differences between the two kinds of self focus, which can be used to reasonably draw out the role of mindfulness.

5. On the third page, I suggest the author explain the connection between self reference processing and self focus.

6. On page 4, I suggest the author introduce the role of self compassion in improving ToM.

7. On page 4, I suggest the author explain the distinction between reflective self-focus and mindful self-focus.

8. On page 5, I suggest that the author explain the reasons for choosing mindfulness before putting forward the hypothesis.

9. On page 6, I suggest the author answer: for subjects from different countries, do the experimental materials and procedures unify the language used? If not unified, is there any deviation in the translation between different languages of the same material?

10. On page 6, I suggest the author briefly describe the classification standard of the code to prove its reliability.

Reviewer #2: In this paper, the authors examined the whether the association between self-focus and theory of mind is moderated by trait mindfulness in a sample with both English and German speakers. They found a small but robust self-focus × mindfulness interaction effect on ToM, such that there was a significant positive association between self-focus and ToM for more mindful individuals and no significant association for less mindful individuals.

The logic of the questions examined is generally well-argued and the paper is overall well-written. I have several comments listed below:

1. The completion time for the experiments varied substantially within the sample, which could influence the results. It is important to include this as a covariate in the analyses. I recommend the authors rerun their analyses by including this as a covariate.

2. Figure 1 is not intuitive. If a moderation was done, please plot the interaction effect. With self-focus on X axis, ToM on Y axis, and mindfulness as the moderator with three levels (e.g., +1, -1SD, mean).

3. Table 3, please move the labels of the variables to the left side rather than having it on the right side. Also, take a look at the 95% CI labels, it’s partially cutting off.

4. Relatedly, it is unclear to me why didn’t the authors just show the “controls model”. From my reading of the paper, the most central part of the analyses, is examining whether or not mindfulness moderates the association between self-focus and ToM, after controlling for all potential covariates that may contribute to the associations. This question can be fully addressed with the “controls model”, which in fact should be the main model. I can potentially see why the authors may want to include the interaction effects model without covariates, but the main effects model is not informative and unnecessary. Instead, I suggest Table 3 should be structured as “without covariates” and “with covariates”.

5. The discussion section needs some work: I suggest the authors broadly going over their research questions and results in the first paragraph, rather than directly jumping into results without introducing the question.

6. PLOS authors have the option to publish the peer review history of their article (what does this mean?). If published, this will include your full peer review and any attached files.

Reviewer #1: No

Reviewer #2: No

---

## [Author Response · Author response to Decision Letter 0]

21 Sep 2022

Response Letter

Note: a. = answer

Reviewer #1: 

In the study of the relationship between self focus and theory of mind (ToM), it is theoretically feasible to include mindfulness level as a regulating variable. The following problems exist:

a. We would like to thank the Reviewer #1 for his helpful suggestions – particularly with regard to finding a clearer language and improving the accessibility of the text for the reader!

1. The measured ToM in this paper is obtained through the coding of the experimenter, and the individual judgment standard of the coder will have an unstable impact on the results.

a. We believe that this might be a misunderstanding: The ToM measure is not obtained through the experimenter’s subjective coding. Instead, we used the DMASC to measure ToM, which is an objective multiple-choice task. We now make this clearer on page 7:

Throughout a 15-minute short movie, the DMASC-MC requires participants to answer 44 items on the thoughts, feelings, and intentions of four characters who spend an evening together (e. g. “Why did Michael say that?”). Each time, participants selected one from four multiple choice options to indicate what they think was true, which was coded as based on the DMASC-MC as “mentalized appropriately”, “too much”, “too little”, or “not at all”.

2. Among the subjects participating in the experiment, a small number of subjects complete the cycle too long, which will lead to different psychological states of the same subject at different times, which will affect the experimental results (on page 8).

a. Thank you for this remark, we have now added participation duration as a covariate and updated the table and manuscript accordingly. This change was also suggested by the other reviewer. Please note that participation duration has no significant impact so that the other results remain largely the same (cf. table page 12 and our hypothesis on page 5).

Study participation duration may introduce noise to the data but should not significantly affect ToM performance in a particular direction or affect its relation to self-focus and mindfulness as both constructs are rather traits than states according to the understanding underlying the used measurements (compare Materials).

Also note that model order has changed in response to the other reviewer’s 4th comment.

3. Before putting forward the hypothesis, I suggest the author introduce the background or reason for choosing mindfulness as the regulating variable? (In abstract)

a. We followed this suggestion and changed the abstract accordingly (page 1):

Is thinking about oneself helpful or harmful for understanding other people? The answer might depend on how a person thinks about themself. Mindfulness is one prominent construct that seems to affect the quality and content of a person’s thoughts about themselves in the world. Thus, we hypothesize that the relationship between self-focus and Theory of Mind (ToM) is moderated by mindfulness.

4. On the second page, I suggest the author outline the differences between the two kinds of self focus, which can be used to reasonably draw out the role of mindfulness.

a. Unfortunately, we are not quite sure, which “two kinds of self focus” the reviewer is referring to. Our understanding of the literature is not that there are two kinds of self-focus but two theories – objective self-awareness theory and the meta-construct model (p. 2) – about how self-focus affects our thinking. To make this clear, we added the following clarification on page 2:

Intuitively, the tendency to think about oneself should bias inferences about other people in an egocentric manner (Fenigstein & Abrams, 1993). Objective self-awareness theory, however, posits that self-focus actually reduces this bias because it suggests self-focus is taking a third-person perspective at oneself (Duval & Wicklung, 1972; Hass & Eisenstadt, 1990; Silvia & Duval, 2001). A third possibility is that self-focus can be helpful or harmful depending on how one thinks about oneself.

If the “two kinds” refers to “reflective and mindful self-focus” please compare our answer to your 7th suggestion below.

5. On the third page, I suggest the author explain the connection between self reference processing and self focus.

a. We now clarify the relation (or slight difference) between the two concepts on page 3:

However, the study investigated the role of self-referential processing which concerns the superior recall of information that has previously been related to oneself as compared to information that has not been related to oneself (Rogers, Kuiper, & Kirker, 1977). In contrast, self-focus is about the act and tendency to relate information to oneself in the first place. As such, self-focus is a prerequisite of self-referential processing.

6. On page 4, I suggest the author introduce the role of self compassion in improving ToM.

a. Thank you for raising this point, however, we would like to decline this suggestion for the following reason: We list and reference self-compassion as just one of many reasons that have been put forward by other researchers to explain how mindfulness might positively affect social cognitive abilities. We do not necessarily think it is the actual mechanisms at work here. While one reading of “mindfulness” is related to self-compassion (e.g. when researching meditation intervention), our reading of mindfulness is in line with our measure of mindfulness (MAAS) as being about “an awareness in and of the present moment” independent of the notion of self-compassion. Thus, we do not think going deeper into the matter of self-compassion strengthens the argument we are putting forward but might be misleading instead.

7. On page 4, I suggest the author explain the distinction between reflective self-focus and mindful self-focus.

a. Thank you for bringing up this point. We intended that “reflective and mindful” be read as synonyms since related working definitions and measurements used in the cited literature largely overlap. It mostly seems to be a shift in the language used over time. Where previous research used “reflective” more recent research uses the term “mindful”. To minimize confusion, it now reads “reflective or mindful” where necessary, i.e. on page 2 and 4:

Indeed, previous research has shown that reflective or mindful self-focus has many benefits for psychological functioning making mindfulness a likely moderator for our hypothesis (Huffziger & Kuehner, 2009; McKie, Askew, & Dudley, 2017; Sauer & Baer, 2012; Trapnell & Campbell, 1999; Watkins, 2008; Watkins & Teasdale, 2004).

A range of distinct adaptive properties have been discerned for reflective or mindful self-focus on mood and psychological functioning (Huffziger & Kuehner, 2009; Sauer & Baer, 2012; Trapnell & Campbell, 1999; E. Watkins & Teasdale, 2004).

8. On page 5, I suggest that the author explain the reasons for choosing mindfulness before putting forward the hypothesis.

a. We followed your suggestion adding the explanation on page 4:

We chose mindfulness as a likely moderator for self-focus because, by definition, mindfulness shapes how we relate to ourselves in the world and thus should determine the content or quality of our self-focus, for example, as being directed at one’s present thoughts and feelings and being non-judgmental about them. A range of distinct adaptive properties have been discerned for reflective or mindful self-focus on mood and psychological functioning (Huffziger & Kuehner, 2009; Sauer & Baer, 2012; Trapnell & Campbell, 1999; E. Watkins & Teasdale, 2004). For example, …

9. On page 6, I suggest the author answer: for subjects from different countries, do the experimental materials and procedures unify the language used? If not unified, is there any deviation in the translation between different languages of the same material?

a. We control for both language and language nativity in our analysis. The results suggest that native speakers performed better on the ToM task than non-native speakers, which makes sense as the outcome requires participants to follow a couple of conversations. The results also suggest that participants participating in English performed better, which may be explained by the fact that the video material for the ToM task was originally recorded in English, while the German version is dubbed leading to a mismatch of audio and visual information that might negatively affect the ToM task performance. All other evaluated experimental materials were validated in German and English (as evidenced by our citations) except for the SFSC, whose items were translated into German by an expert speaker of both languages for this study. The simplicity of the SFSC items (e.g. original English “My father…”, “I wonder…”) makes it reasonable to assume that their meaning did not deviate significantly across languages.

10. On page 6, I suggest the author briefly describe the classification standard of the code to prove its reliability.

a. Thank you for this remark, we added the following description on page 6:

Self-focus was measured with the self-focus sentence completion task (SFSC; Exner, 1973), which requires subjects to finish 30 open-ended sentences prompting responses concerning themselves or others, e. g. “If only I could …”. Each half-sentence response was coded by three raters according to the coding scheme suggested by Exner across his six categories: “egocentric” (self-focused, e.g. “… live my life freely.”), “egocentric and negative” (e.g. “… end my life.”), “allocentric” (other-focused, e.g. “… help my sister.”), “allocentric and emotional” (e.g. “… stop hating my father for what he has done.”), “both” (self- and other-focused, e.g. “… repair my relationship with my mother.”), and “other” for answers that do not relate to a person or are too short (e.g. “… fly”). We followed modern research practice and evaluated the SFSC based on the count of “egocentric” and the “egocentric and negative” responses (Philippi et al., 2018; Philippi, Dahl, Jany, & Bruce, 2019; Woodruff-Borden et al., 2001). We divided their sum by the number of raters and the number of SFSC items to get a ratio of self-focus that is as unbiased as possible by the coding variability of an individual rater. The three raters were psychology students previously trained on a pilot sample (N = 73). Interrater reliability was Fleiss’ κ = 0.695 (p < 0.001, 95% CI [0.692; 0.697]; (Fleiss & Cohen, 1973).

Reviewer #2:

In this paper, the authors examined the whether the association between self-focus and theory of mind is moderated by trait mindfulness in a sample with both English and German speakers. They found a small but robust self-focus × mindfulness interaction effect on ToM, such that there was a significant positive association between self-focus and ToM for more mindful individuals and no significant association for less mindful individuals.

The logic of the questions examined is generally well-argued and the paper is overall well-written. I have several comments listed below:

a. Thank you very much for the positive assessment of our manuscript! Also, we would like to thank the Reviewer #2 for his helpful suggestions – particularly with regard to adhering more to common standards presenting the relevant tables and figures.

1. The completion time for the experiments varied substantially within the sample, which could influence the results. It is important to include this as a covariate in the analyses. I recommend the authors rerun their analyses by including this as a covariate.

a. Thank you for this remark, we have now added participation duration as a covariate and updated the table and manuscript accordingly. This change was also suggested by the other reviewer. Please note that participation duration has no significant impact so that the other results remain largely the same (cf. table page 12 and our hypothesis on page 5).

Study participation duration may introduce noise to the data but should not significantly affect ToM performance in a particular direction or affect its relation to self-focus and mindfulness as both constructs are rather traits than states according to the understanding underlying the used measurements (compare Materials).

2. Figure 1 is not intuitive. If a moderation was done, please plot the interaction effect. With self-focus on X axis, ToM on Y axis, and mindfulness as the moderator with three levels (e.g., +1, -1SD, mean).

a. As suggested, we added a second figure showing the interaction effect in the–admittedly–more common way of presenting moderation effects but also kept Figure 1 because in contrast to Figure 2 it retains the continuity of the continuous moderation effect. Additionally, we slightly changed Figure 1 from highlighting quartiles to SD and Mean to be aligned with Figure 2 both of which can be found on page 13.

3. Table 3, please move the labels of the variables to the left side rather than having it on the right side. Also, take a look at the 95% CI labels, it’s partially cutting off. Relatedly, it is unclear to me why didn’t the authors just show the “controls model”. From my reading of the paper, the most central part of the analyses, is examining whether or not mindfulness moderates the association between self-focus and ToM, after controlling for all potential covariates that may contribute to the associations. This question can be fully addressed with the “controls model”, which in fact should be the main model. I can potentially see why the authors may want to include the interaction effects model without covariates, but the main effects model is not informative and unnecessary. Instead, I suggest Table 3 should be structured as “without covariates” and “with covariates”.

a. Thank you for this remark. We now include three models: a covariates model, an interaction model, and a joined model. We include all three, because we believe there are two schools of thought regarding covariates: One interested in whether the hypothesized effects remain even after adding the covariates, which is what we did, and one school interested in whether the hypothesized effects add something after other reasonable predictors have been accounted for. Two differently flavored interpretations. By incorporating your suggestion, we address both points of view and might provide a more complete picture of which variables do or do not take away explained variance from each other. The new table can be found on page 12.

4. The discussion section needs some work: I suggest the authors broadly going over their research questions and results in the first paragraph, rather than directly jumping into results without introducing the question.

a. We added a new opening paragraph to the discussion on page 14:

We set out asking whether thinking about oneself is helpful or harmful for ToM performance. Reviewing the sparse and mixed literature, we found Ingram’s (1990) theory most compelling that the answer may depend on the specific content or quality a person’s self-focus can take. We considered mindfulness a psychological construct that should affect the content or quality of self-focus because mindfulness specifies towards what and how a person focuses their attention. Thus, we explored the idea whether the relationship between self-focus and ToM performance is moderated by mindfulness. Overall, our results are in line with Ingram’s idea finding different effects of self-focus on ToM performance depending on a person’s level of mindfulness.

---

## [Editor Report · Decision Letter 1]

26 Sep 2022

PONE-D-22-04751R1Mindful self-focus–an interaction affecting Theory of Mind?PLOS ONE

Dear Dr. Wundrack,

Thank you for submitting your manuscript to PLOS ONE. After careful consideration, we feel that it has merit but does not fully meet PLOS ONE’s publication criteria as it currently stands. Therefore, we invite you to submit a revised version of the manuscript that addresses the points raised during the review process. Please ensure that your decision is justified on PLOS ONE’s publication criteria and not, for example, on novelty or perceived impact.

We look forward to receiving your revised manuscript.

Kind regards,

Yi-Yuan Tang

Academic Editor

PLOS ONE
---

## [Author Response · Author response to Decision Letter 1]

29 Sep 2022

Dear Yi-Yuan Tang,

we have now uploaded the figures to PACE as requested. Please excuse for not doing so earlier. We have also replaced to figures in the uploaded documents "manuscript" and "manuscript with track changes". We hope our manuscript is now up to PLOS ONE standards.

Kind regards, 

the authors

---

## [Decision Letter · Decision Letter 2]

12 Dec 2022

Mindful self-focus–an interaction affecting Theory of Mind?

PONE-D-22-04751R2

Dear Dr. Wundrack,

We’re pleased to inform you that your manuscript has been judged scientifically suitable for publication and will be formally accepted for publication once it meets all outstanding technical requirements.

Kind regards,

Yi-Yuan Tang

Academic Editor

PLOS ONE

Additional Editor Comments (optional):

Reviewers' comments:

Reviewer's Responses to Questions

**Comments to the Author**

1. If the authors have adequately addressed your comments raised in a previous round of review and you feel that this manuscript is now acceptable for publication, you may indicate that here to bypass the “Comments to the Author” section, enter your conflict of interest statement in the “Confidential to Editor” section, and submit your "Accept" recommendation.

Reviewer #2: All comments have been addressed

2. Is the manuscript technically sound, and do the data support the conclusions?

Reviewer #2: Yes

3. Has the statistical analysis been performed appropriately and rigorously? 

Reviewer #2: Yes

4. Have the authors made all data underlying the findings in their manuscript fully available?

Reviewer #2: Yes

5. Is the manuscript presented in an intelligible fashion and written in standard English?

Reviewer #2: Yes

6. Review Comments to the Author

Reviewer #2: I am satisfied with this revision and I applaud the authors for making the necessary changes to their manuscript. Thank you.

7. PLOS authors have the option to publish the peer review history of their article (what does this mean?). If published, this will include your full peer review and any attached files.

Reviewer #2: No

---

## [Editor Report · Acceptance letter]

16 Dec 2022

PONE-D-22-04751R2 

Mindful self-focus–an interaction affecting Theory of Mind? 

Dear Dr. Wundrack:

I'm pleased to inform you that your manuscript has been deemed suitable for publication in PLOS ONE. Congratulations! Your manuscript is now with our production department. 

Kind regards, 

on behalf of

Dr. Yi-Yuan Tang 

Academic Editor

PLOS ONE